Ticagrelor alleviates pyroptosis of myocardial ischemia reperfusion-induced acute lung injury in rats: a preliminary study

Dai Yi-Ning 1 2
Wang Li-Tao 1 2
Zhang Ye-Shen 1 2
Xue Ling 1 2
He Peng-Cheng 1 2
Tan Ning 1 2 gdtanning@126.com
Liu Yuan-Hui 1 2 liuyuanhui@gdph.org.cn
1 Department of Cardiology, Guangdong Cardiovascular Institute, Guangdong Provincial People’s Hospital (Guangdong Academy of Medical Sciences), Southern Medical University , Guangzhou , China
2 Guangdong Provincial Key Laboratory of Coronary Heart Disease Prevention, Guangdong Provincial People’s Hospital , Guangzhou , China
Oliveira Sonia
Electronic publication date: 2024 Jan 4
Publication date: 2024
Volume: 12
Electronic Location ID: e16613
Received 2023 Mar 21; Accepted 2023 Nov 15
Copyright: © 2024 Dai et al.
Copyright year: 2024
Copyright holder: Dai et al.
License: This is an open access article distributed under the terms of the Creative Commons Attribution License, which permits unrestricted use, distribution, reproduction and adaptation in any medium and for any purpose provided that it is properly attributed. For attribution, the original author(s), title, publication source (PeerJ) and either DOI or URL of the article must be cited.
License URL: https://creativecommons.org/licenses/by/4.0/

Keywords: Myocardial ischemia-reperfusion injury, Acute lung injury, Pyroptosis, NLRP3, Ticagrelor

Funding: Shuangqing Talent Program Project of Guangdong Provincial People’s Hospital KJ012019095 Central Universities 2022ZYGXZR039 Shuangqing Talent Program Project of Guangdong Provincial People’s Hospital KJ012019084 Science and Technology Planning Project of Guangzhou City 2023A04J0486 This work was supported by the Shuangqing Talent Program Project of Guangdong Provincial People’s Hospital [Grant No. KJ012019095 to Y.H.L and Grant No. KJ012019084 to H.P.C] and the Fundamental Research Funds for the Central Universities [Grant No. 2022ZYGXZR039 to Y.H.L]. The Shuangqing Talent Program Project of Guangdong Provincial People’s Hospital [Grant No. KJ012019084 to H.P.C] and the Science and Technology Planning Project of Guangzhou City [Grant No. 2023A04J0486 to Y.N.D] supported the APC of this article. The funders had no role in study design, data collection and analysis, decision to publish, or preparation of the manuscript.

==============================
Pulmonary infection is highly prevalent in patients with acute myocardial infarction undergoing percutaneous coronary intervention. However, the potential mechanism is not well characterized. Myocardial ischemia-reperfusion injury (MIRI) induces acute lung injury (ALI) related to pulmonary infection and inflammation. Recent studies have shown that pyroptosis mediates ALI in several human respiratory diseases. It is not known whether MIRI induces pyroptosis in the lungs. Furthermore, ticagrelor is a clinically approved anti-platelet drug that reduces ALI and inhibits the expression levels of several pyroptosis-associated proteins, but the effects of ticagrelor on MIRI-induced ALI have not been reported. Therefore, we investigated whether ticagrelor alleviated ALI in the rat MIRI model, and its effects on pyroptosis in the lungs. Sprague-Dawley rats were randomly divided into four groups: control, MIRI, MIRI plus low ticagrelor (30 mg/kg), and MIRI plus high ticagrelor (100 mg/kg). Hematoxylin and Eosin (HE) staining was performed on the lung sections, and the HE scores were calculated to determine the extent of lung pathology. The wet-to-dry ratio of the lung tissues were also determined. The expression levels of pyroptosis-related proteins such as NLRP3, ASC, and Cleaved caspase-1 were estimated in the lung tissues using the western blot. ELISA was used to estimate the IL-1β levels in the lungs. Immunohistochemistry was performed to determine the levels of MPO-positive neutrophils as well as the total NLRP3-positive and Cleaved caspase-1-positive areas in the lung tissues. The lung tissues from the MIRI group rats showed significantly higher HE score, wet-to-dry ratio, and the MPO-positive area compared to the control group, but these effects were attenuated by pre-treatment with ticagrelor. Furthermore, lung tissues of the MIRI group rats showed significantly higher expression levels of pyroptosis-associated proteins, including NLRP3 (2.1-fold, P < 0.05), ASC (3.0-fold, P < 0.01), and Cleaved caspase-1 (9.0-fold, P < 0.01). Pre-treatment with the high-dose of ticagrelor suppressed MIRI-induced upregulation of NLRP3 (0.46-fold, P < 0.05), ASC (0.64-fold, P < 0.01), and Cleaved caspase-1 (0.80-fold, P < 0.01). Immunohistochemistry results also confirmed that pre-treatment with ticagrelor suppressed MIRI-induced upregulation of pyroptosis in the lungs. In summary, our data demonstrated that MIRI induced ALI and upregulated pyroptosis in the rat lung tissues. Pre-treatment with ticagrelor attenuated these effects.

Introduction

Globally, acute myocardial infarction (AMI) is the leading cause of mortality (Anderson & Morrow, 2017; Geng et al., 2022; Shaheen, Helal & Anan, 2021). Pulmonary infection is highly prevalent in patients with AMI undergoing percutaneous coronary intervention (PCI) and is associated with elevated mortality rates (Piccaro de Oliveira et al., 2016; Putot et al., 2019; Truffa et al., 2012). However, the cause of lung injury in patients with AMI undergoing PCI is not known. Moreover, a large number of patients with AMI complicated by cardiogenic shock required mechanical ventilation, indicating the hazards of myocardial ischemia-reperfusion injury (MIRI) for acute lung injury (ALI) (Vallabhajosyula et al., 2019).

Studies have reported that MIRI induced ALI and activated apoptosis pathway in the lungs (Gao et al., 2017; Sezen et al., 2018). ALI is characterized by the infiltration of neutrophils, diffuse alveolar damage, and pneumonia (Hughes & Beasley, 2017). Pyroptosis is a novel mechanism of programmed cell death that is triggered by proinflammatory signals and is characterized by the formation of pores in the plasma membrane (Wen et al., 2022). Canonical pyroptosis is induced by the NLRP3 inflammasome-mediated activation of caspase-1, which promotes pore formation in the plasma membrane and maturation of IL-1β and IL-18 (Zhao et al., 2022). Pyroptosis is a potential therapeutic target for several inflammatory diseases, including AMI (Toldo et al., 2018). Pyroptosis of the pulmonary epithelial cells, neutrophils, and macrophages enhances inflammation and promotes development of ALI (Sun & Li, 2022). NLRP3 inflammasome plays a central role in pyroptosis, and its dysregulation is associated with various infectious respiratory diseases (Brusselle et al., 2014). However, it is not clear whether NLRP3 inflammasome-mediated pyroptosis plays a role in ALI after AMI.

Ticagrelor is a potent anti-platelet drug for patients with AMI (Wallentin et al., 2009). Ticagrelor inhibits platelet aggregation by blocking the binding of ADP to the P2Y12 receptor. Recent studies have shown that ticagrelor might reduce infection occurrences in the clinic, especially for pneumonia (Li et al., 2021; Storey et al., 2014; Varenhorst et al., 2012). Furthermore, ticagrelor alleviates ALI and/or inflammation in vivo against infections, including abdominal sepsis and pneumonia, and in response to myocardial ischemia-reperfusion injury (Fındık et al., 2019; Lancellotti et al., 2019; Rahman et al., 2014). Ticagrelor inhibits several pyroptosis-associated proteins, including NLRP3, adaptor protein apoptosis-associated speck-like protein containing a caspase recruitment domain (CARD) (ASC), and caspase-1 in cells as well as several experimental models (Birnbaum et al., 2016; Chen et al., 2020; Huang et al., 2020; Penna et al., 2020). However, it is not clear whether ticagrelor alleviates MIRI-associated ALI by suppressing pyroptosis.

Therefore, in this study, we investigated whether ticagrelor inhibited MIRI-induced ALI and pyroptosis, in the rat model. Since previous studies have demonstrated that pyroptosis is a key process in the inflammatory and pathological processes of infectious diseases, our study suggested that pyroptosis is a potential mechanism that enhances the prevalence of pulmonary infections in patients with AMI undergoing percutaneous coronary intervention. Therefore, pyroptosis is a potential therapeutic target for preventing or treating ALI in these patients. Furthermore, ticagrelor is a widely used anti-platelet drug in patients with AMI. Our data demonstrated that ticagrelor alleviated MIRI-induced pyroptosis and ALI. Therefore, ticagrelor is a promising clinical drug for treating AMI patients with a high-risk of pulmonary infections.

Materials and Methods

Portions of this text were previously published as part of a preprint: https://www.researchsquare.com/article/rs-1550421/v1.

Ethical approval

The animal experiments and protocols were performed in compliance with relevant institutional and national regulations for the protection of animals, and all the necessary efforts were made to reduce the number of animals for the experiments and limit their suffering. The animal protocols were approved by the Research Ethics Committee of the Guangdong Provincial People’s Hospital (Approval No. KY-Z-2021-2137-01).

Animals

A total of 8-to-12-week-old pathogen-free male Sprague-Dawley (SD) rats weighing 250–300 g were purchased from the Guangdong experimental animal center in China. The rats were provided with food and water ad libitum and maintained under sterile conditions.

Ticagrelor

Ticagrelor powder (Cat# SML2482; Sigma, Burlington, MA, USA) was stored at −20 °C. The purity of ticagrelor was ≥98% with a solubility of 2 mg/ml in dimethyl sulfoxide (DMSO). The time to peak effect of ticagrelor was 2 h; the time of maximum concentration for the active metabolite of ticagrelor was 2–3 h; and the plasma half-life of ticagrelor was 9–12 h (Schilling, Dingemanse & Ufer, 2020).

Experimental protocol

The rats were randomly divided into the following four groups (n = 6 each): control, MIRI, MIRI plus low ticagrelor (30 mg/kg), and MIRI plus high ticagrelor (100 mg/kg). Ticagrelor was dissolved in saline and then administered 2 h before MIR surgery via intragastric gavage (Rahman et al., 2014). Rats in the control and the MIRI groups were administered equal volumes of saline.

The rats were anesthetized via intraperitoneal injection of pentobarbital sodium (50 mg/kg). Then, the hearts were surgically exposed. The depth of anesthesia was confirmed by testing the corneal reflex and observing the changes in the pupil size and breathing. The anesthesia effects were monitored using the electrocardiogram. When the rats seemed to be coming out of anesthesia based on their increased reaction and breathing during surgery, they were administered additional pentobarbital sodium.

Myocardial ischemia was induced by ligating the left anterior descending coronary artery with a 7/0 nylon suture at 2 mm below the left auricle. The rats were immediately covered with a heating blanket after the artery ligation and the core body temperature was monitored using a non-contact infrared thermometer. The ligation was performed for 30 min. Then, the rats were subjected to reperfusion for 120 min. Pentobarbital sodium were used for anesthesia at the end of procedure, after confirmed the depth of anesthesia, rats were euthanized with bleeding the abdominal aorta. The success of MIRI surgery was analyzed by estimating the serum levels of creative kinase MB (CKMB, Cat. No. C060-e) and lactate dehydrogenase (LDH; Cat. No. C018-e) in an automatic biochemistry analyzer using the corresponding commercially available kits from Changchun Huili Biological Technology Co. Ltd. (Changchun, China). The blood samples and tissues were pre-treated and stored at −80 °C.

Estimation of the wet to dry weight ratio

The left lung was used to calculate the wet-to-dry weight (W/D) ratio. The wet weight of the lung was measured immediately after excision. The lung tissue was then dried in an oven at 65 °C for 72 h and then weighed to determine the dry weight.

Hematoxylin & Eosin staining and evaluation of lung injury

The right lung tissue was fixed with 10% paraformaldehyde for 48 h, embedded in paraffin, and the blocks were cut into 4 μm thick sections. The paraffin embedded sections were processed and stained with H&E, and photographed under an optical microscope. The lung damage was evaluated independently by two pathologists who were blinded to the nature of samples. An Hematoxylin & Eosin (HE) score was estimated for all the samples in three random areas of each section based on the extent of hemorrhage, edema, congestion, and inflammation. Based on the HE scores, the extent of lung damage was assessed as follows: zero, no obvious damage; one, slight damage; two, moderate damage; three, severe damage; and four, maximal damage.

Western blotting and ELISA

Total protein extracts of lung tissue samples were prepared with the RIPA buffer. The protein concentration of the samples was measured using the BCA assay. Equal amounts of lung tissue protein samples (20–30 μg) were separated on a SDS-PAGE and then transferred to a nitrocellulose membrane (Gelman Laboratory, Ann Arbor, MI, USA). The membranes were blocked with 5% BSA for 1 h. Then, the blots were incubated at 4 °C overnight with the following primary antibodies from Abcam (Cambridge, UK): anti-NLRP3 (1:500; Cat# ab214185), anti-Cleaved caspase-1 (1:2,000, Cat# ab179515), and anti-ASC (1:1,000, Cat# ab18193); and anti-β-actin (1:2,000, Cat# ab8227; internal control). Subsequently, the membranes were incubated with the goat anti-rabbit IgG antibody (1:5,000, Cat# ab6721) at room temperature for 1 h. After the washing steps, blots were developed using the ECL reagent (Merck Millipore, Hayward, CA, USA). The immunoreactive bands were visualized using a fluorescence band detection system. The qualification of each protein of western blotting were following: firstly, pictures were adjusted to 8-bit image and selected the associated band (contain four groups). Secondly, created plot lanes and used long strings to separate each group at the bottom of the tough. Third, measured the area of each group. Band densities were calculated using the Image J software (National Institutes of Health, Bethesda, MD, USA). IL-1β levels in the lung tissue protein samples were determined using the ELISA kit (MSKBIO, Wuhan, China) according to the manufacturer’s instructions.

Immunohistochemistry

Immunohistochemical (IHC) staining of the paraffin-embedded lung tissues were performed according to standard protocols. The processed sections were incubated overnight at 4 °C with the following primary antibodies: anti-myeloperoxidase (MPO) (1:500; Cat# GB11224; Servicebio, Wuhan, China), anti-NLRP3 (1:400; Cat# ab214185; Abcam, Cambridge, UK), and anti-Cleaved caspase-1 (1:200, Cat# AF4022; Affinity, Cincinnati, OH, USA). Subsequently, the sections were incubated with the FITC-labeled secondary antibodies for 1 h at room temperature in the dark. Then, the IHC-stained sections were photographed using an optical microscope. The percentage of positive staining in each sample was estimated using the Image J software, by following steps: firstly, switched pictures to 8-bit image and also adjusted the contrast if necessary. Secondly, compared and adjusted the threshold of the pictures to make sure the positive areas were colored by red. Third, measure the areas of red color.

Statistical analysis

The data are represented as mean ± SEM. One-way ANOVA followed by the LSD test was used to compare groups for data showing Gaussian distribution and homogeneity of variance. Otherwise, Mann-Whitney Test followed by Dunnett’s T3 test was used. P values were estimated according to the LSD or Dunnett’s T3 tests. SPSS software version 24.0 (IBM Corp., Chicago, IL, USA) was used for statistical analyses. Two tailed P-value < 0.05 was considered as statistically significant.

Results

Ticagrelor attenuates MIRI-induced ALI in rats

The MIRI group showed significantly higher levels of CKMB (2.5-fold; P < 0.01) and LDH (1.7-fold; P < 0.01) compared to the control group (Figs. 1A and 1B). This confirmed that MIRI was successfully established. However, pre-treatment with high or low doses of ticagrelor attenuated MIRI-induced elevation of serum CKMB and LDH levels (P < 0.01 compared with the MIRI group) (Figs. 1A and 1B).

Figure 1 The expression levels of myocardial enzymes in the serum and the histopathological analysis of ALI in the MIRI and ticagrelor-treatment groups.

(A and B) Serum levels of CKMB and LDH in the control, MIRI, MIRI plus low ticagrelor, and MIRI plus high ticagrelor groups of rats (n = 6 each). **P < 0.01 compared with the MIRI group. (C) Wet to dry weight (W/D) ratio of the lung tissues from the control, MIRI, MIRI plus low ticagrelor, and MIRI plus high ticagrelor groups of rats (n = 6 each). **P < 0.01 compared with the MIRI group. (D) HE scores for the control, MIRI, MIRI plus low ticagrelor, and MIRI plus high ticagrelor groups of rats (n = 6 each). As shown, HE scores were high in the MIRI group but reduced in groups pre-treated with ticagrelor (n = 6). **P < 0.01 compared with the MIRI group. (E) Representative images show H&E-stained lung tissue sections from the control, MIRI, MIRI plus low ticagrelor, and MIRI plus high ticagrelor groups of rats (n = 6 each). Lung tissues in the MIRI group show neutrophil exudation, alveolar wall thickening, and hemorrhage, but these effects are attenuated by ticagrelor. Magnification, 200×; A: alveoli, I: inflammation, H: hemorrhage. (F and G) MPO staining results in the lung tissues of the control, MIRI, MIRI plus low ticagrelor, and MIRI plus high ticagrelor groups of rats (n = 6 each) to detect neutrophils. Lung tissues of the MIRI group show higher number of MPO-positive neutrophils, but their numbers are significantly lower or absent in the ticagrelor pre-treatment groups. Magnification, 200×; arrow denotes MPO-positive cells; **P < 0.01 compared with the MIRI group. The statistical results are represented as mean ± SEM. LT: low ticagrelor; HT: high ticagrelor.

The W/D ratio of lungs was significantly higher in the MIRI group rats compared to the control group rats (P < 0.01). This suggested pulmonary edema in the MIRI group rats. However, W/D ratio of lungs was significantly reduced in the high- and low-dose ticagrelor groups (P < 0.01 compared with MIRI group) (Fig. 1C). H&E staining results demonstrated significant lung injury in the MIRI group rats, including hemorrhage, edema, congestion, and inflammation, and this was observed in the HE-scores (Figs. 1D and 1E). HE scores were significantly higher in the MIRI group compared with the control group (P < 0.01 with control group), but the HE scores for the low- and high-dose ticagrelor were lower compared to the MIRI group (P < 0.01 compared with the MIRI group) (Fig. 1D).

MPO staining results showed significantly high infiltration of the MPO-positive neutrophils cells in the lung tissues from the MIRI group rats compared with the control group, but these effects were abrogated in the low- and high-dose ticagrelor groups (both P < 0.01) (Figs. 1F and 1G). Together, these results confirmed that MIRI induced ALI in the rats. However, pre-treatment with ticagrelor attenuated the pathological changes of ALI after MIRI.

Ticagrelor attenuates MIRI-induced pyroptosis in the lungs

Western blot analysis showed significantly high expression of pyroptosis-associated proteins such as NLRP3 (2.1-fold, P < 0.05), ASC (3.0-fold, P < 0.01), and Cleaved caspase-1 (9.0-fold, P < 0.01) in the lung tissues of the MIRI group compared to those from the control group (Figs. 2A–2D). ELISA assay results showed significantly high levels of IL-1β in the lung tissues of the MIRI group compared with the control group (1.5-fold; P < 0.01) (Fig. 2E).

Figure 2 Western blot analysis of the expression levels of pyroptosis-associated proteins in the rat lung tissues of the MIRI group and ticagrelor-treatment group.

(A–D) Western blot analysis shows the expression levels of NLRP3, ASC, and Cleaved caspase-1 in the lung tissues of the control, MIRI, MIRI plus low ticagrelor, and MIRI plus high ticagrelor groups of rats (n = 6 each). *P < 0.05, **P < 0.01 compared with the MIRI group. (E) ELISA assay results show the levels of IL-1β in the lung tissues of the control, MIRI, MIRI plus low ticagrelor, and MIRI plus high ticagrelor groups of rats (n = 6 each). **P < 0.01 compared with MIRI group. The results are represented as mean ± SEM. LT: low ticagrelor; HT: high ticagrelor.

However, lung tissues of rats in the high-dose ticagrelor group showed significantly lower levels of NLRP3 (0.46-fold reduction, P < 0.05), ASC (0.64-fold reduction, P < 0.01), Cleaved caspase-1 (0.80-fold reduction, P < 0.01), and IL-1β (0.09-fold reduction, P < 0.01) compared with the MIRI group (Figs. 2A–2E). Furthermore, lung tissues of rats in the low-dose ticagrelor group also demonstrated reduced levels of NLRP3 (0.22-fold reduction), ASC (0.28-fold reduction), Cleaved caspase-1 (0.42-fold reduction), and IL-1β (0.04-fold reduction) compared with the MIRI group, but the differences were not statistically significant (Figs. 2A–2E). These results demonstrated that MIRI induced pyroptosis in the lungs, but these effects were abrogated by pre-treatment with a high dose of ticagrelor.

Immunohistochemical analysis demonstrates that ticagrelor attenuates MIRI-induced pyroptosis in the lungs

IHC assay demonstrated that the NLRP3+ (P < 0.05) and Cleaved caspase-1+ (P < 0.01) area was significantly increased in the lung tissues of the MIRI group compared to the control group (Figs. 3A–3D). However, the NLRP3+ area was significantly reduced in both the high- and low-dose ticagrelor groups (P < 0.05) compared with the MIRI group (Figs. 3A and 3C). Moreover, compared with the MIRI group, the Cleaved caspase-1+ area was significantly reduced in the lung tissues of the high ticagrelor group (P < 0.01), and was reduced but without statistical significance (P > 0.05) in the lung tissues of the low ticagrelor group (Figs. 3B and 3D). These data confirmed that pyroptosis-related proteins were upregulated in the lung tissues of the MIRI group, but these effects were abrogated by pre-treatment with ticagrelor.

Figure 3 IHC analysis of pyroptosis-associated proteins in the rat lung tissues of MIRI and ticagrelor-treatment groups.

(A and B) IHC staining of (A) NLRP3 and (B) Cleaved caspase-1 in the lung tissue sections of the control, MIRI, MIRI plus low ticagrelor, and MIRI plus high ticagrelor groups of rats (n = 6 each). The arrow in (A) denotes the NLRP3-positive cells and the arrow in (B) denotes the Cleaved caspase-1-positive cells. (C and D) The total NLRP3-positive and Cleaved caspase-1-positive area was higher in the MIRI group but was decreased when pre-treated with ticagrelor. Magnification, 200×; n = 6; *P < 0.05 and **P < 0.01 compared with the MIRI group. The results are represented as mean ± SEM. LT: low ticagrelor; HT: high ticagrelor.

Discussion

The present study demonstrated that ALI happened, and pyroptosis-associated proteins such as NLRP3, ASC, and Cleaved caspase-1 were significantly increased in the lung tissues of rats after MIRI, but these effects were attenuated by pre-treatment with ticagrelor. Several studies in the rat models have shown that MIRI induced ALI (Alkan et al., 2015; Gao et al., 2017; Kip et al., 2015; Sezen et al., 2018). Our results are consistent with these previous reports. Previous studies have shown that lung function and integrity is significantly influenced by MIRI. Clinically, the incidence of pulmonary infections is significantly high in patients with AMI. Therefore, in this study, we investigated MIRI-induced pyroptosis in the lungs. Ischemia-reperfusion injury is associated with increased local expression of pyroptosis-related proteins in the affected organs (Fei et al., 2020; Zhong et al., 2020). However, very few studies have investigated whether ischemia-reperfusion injury-induced pyroptosis in one organ induced acute injury and pyroptosis in another organ. Renal ischemia-reperfusion injury (Liu et al., 2020) and limb ischemia-reperfusion injury (Huang et al., 2021) induced ALI and enhanced pyroptosis. Furthermore, Zhao et al. (2018) reported that renal graft-induced ischemia–reperfusion injury promoted pyroptosis in the remote liver. In the present study, we demonstrated pyroptosis in the lungs after MIRI. This may be due to two plausible mechanisms. Firstly, the circulating cellular debris generated during MIRI is recognized as damage-associated molecular patterns, which activate NLRP3 and the downstream signaling pathways (Toldo & Abbate, 2018). Secondly, MIRI significantly upregulated the secretion of pro-inflammatory cytokines, including TNF, which induced NLRP3-associated pyroptosis (Jorgensen, Rayamajhi & Miao, 2017). Therefore, targeting these mechanisms may reduce MIRI-induced ALI, but further analysis is necessary to determine the underlying molecular mechanisms.

Our findings also confirmed previous reports that ticagrelor attenuated NLRP3 activation in various animal models. The treatment of diabetic ZDF rats with ticagrelor (150 mg/kg/day) for 3 days following MIRI significantly reduced NLRP3 mRNA levels; moreover, ticagrelor reduced caspase-1 expression in the cardiomyocytes (Birnbaum et al., 2016). Furthermore, the treatment of type two diabetes mellitus model mice with 100 mg/kg/day ticagrelor for 12 weeks attenuated the progression of diabetic cardiomyopathy and reduced the expression levels of NLRP3, caspase-1, and GSDMD-N in the myocardium (Chen et al., 2020). Pyroptosis was significantly elevated in the diabetes mellitus model rats compared with the control SD rats (Ding et al., 2019). Therefore, the effects of ticagrelor may be attenuated in the SD rats, and caution should be exercised when extrapolating the data regarding the effects of ticagrelor in the diabetes model rats to the SD rats. Regarding nondiabetic research, the study conducted a MIRI model and showed that ticagrelor administration for 3 days (150 mg/kg daily) reduced the myocardial expression of NLRP3 and protected the heart. These effects disappeared when isolated myocardial cells were treated. This suggested that the myocardial protective effects of ticagrelor were potentially through the platelets rather than direct effects in the cells of the myocardial tissue (Penna et al., 2020). Huang et al. (2020) used the alum-induced peritonitis and lipopolysaccharide-induced sepsis mice models and showed that treatment with 50 mg/kg of ticagrelor inhibited activation of the NLRP3 inflammasome by attenuating the oligomerization of ASC in the macrophages, reduced the inflammatory response, including secretion of IL-1β, and was independent of the P2Y12 signaling pathway. The results of the study by Penna et al. (2020) were partly contradictory with the findings by Huang et al. (2020). These differences may be due to different methodologies used in both studies. The study by Penna et al. (2020) analyzed the myocardium, whereas Huang et al. (2020) analyzed the macrophages, lavage fluid in the peritoneal cavity, and the serum samples. This suggested that ticagrelor targeted diverse pathways that regulated the expression and activity of the NLRP3 inflammasome in different tissues and cells. Therefore, the effects of ticagrelor on different tissues requires in-depth investigations in various experimental models. Our study demonstrated that ticagrelor reduced NLRP3 levels by modulating pyroptosis in the lung tissues of the MIRI model rats. Since we did not find any report that described the effects of ticagrelor on the lung tissues in the MIRI model, we used a low dose of 30 mg/kg ticagrelor and a higher dose of 100 mg/kg ticagrelor for this study. This was based on a previous study that reported inhibition of adenosine diphosphate-dependent platelet aggregation by ≥30 mg/kg ticagrelor, and attenuation of ALI in abdominal sepsis by treatment with 100 mg/kg ticagrelor (Rahman et al., 2014). The ticagrelor doses used in our study were relatively low compared to those used in most of the aforementioned studies.

Our study also has a few limitations. Firstly, we used a previously published protocol for MIRI, which included 120 min ischemia followed by a 30 min reperfusion protocol (Bøtker et al., 2018). We then characterized lung pathology according to a previously published protocol (Gao et al., 2017). However, we did not analyze MIRI-induced lung pyroptosis model at different time points or different ischemia-reperfusion protocol. Secondly, we did not analyze the relationship between lung apoptosis and lung pyroptosis in this study. Thirdly, we established the association between MIRI and lung pyroptosis in this study, but the underlying molecular mechanism requires further investigation.

Conclusions

Our data demonstrated that MIRI induced ALI and upregulated pyroptosis-associated proteins, including NLRP3, ASC, Cleaved caspase-1, and IL-1β in the lung tissues of the MIRI model rats, and these effects were abrogated by pre-treatment with ticagrelor.

Supplemental Information

Supplemental Information 1 Raw pictures of the study.

Click here for additional data file.

Supplemental Information 2 Raw data of the study.

Click here for additional data file.

Supplemental Information 3 Author_Checklist.

Click here for additional data file.

Additional Information and Declarations

Competing Interests

Author Contributions

Animal Ethics

Data Availability

The authors declare that they have no competing interests.

Yi-Ning Dai performed the experiments, analyzed the data, prepared figures and/or tables, authored or reviewed drafts of the article, and approved the final draft.

Li-Tao Wang performed the experiments, authored or reviewed drafts of the article, and approved the final draft.

Ye-Shen Zhang performed the experiments, authored or reviewed drafts of the article, and approved the final draft.

Ling Xue analyzed the data, authored or reviewed drafts of the article, and approved the final draft.

Peng-Cheng He conceived and designed the experiments, authored or reviewed drafts of the article, and approved the final draft.

Ning Tan conceived and designed the experiments, authored or reviewed drafts of the article, and approved the final draft.

Yuan-Hui Liu conceived and designed the experiments, authored or reviewed drafts of the article, and approved the final draft.

The following information was supplied relating to ethical approvals (i.e., approving body and any reference numbers):

All of the experimental procedures was in compliance with the legislation on protecting of animals and was approved by the research ethics committee of Guangdong Provincial People’s Hospital (No. KY-Z-2021-2137-01).

The following information was supplied regarding data availability:

The raw data are available in the Supplemental Files.

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
