# Peer review of "Ticagrelor alleviates pyroptosis of myocardial ischemia reperfusion-induced acute lung injury in rats: a preliminary study"

_PeerJ, doi:10.7717/peerj.16613_

## Round 0.1 · original submission · Major Revisions

All reviewers' comments must be carefully addressed as recommended. Besides, the authors must increase experimental replicates and properly validate the experiments presented.

Reviewer 1 ·

Basic reporting

The language is clear except needs some grammar correction in few places.
The authors had provided sufficient background and hypothesis.
The article is structured appropriately and provided with all necessary information needed.

Experimental design

The aim and scope of the study is appropriate. The information provided in the methods are sufficient.

Validity of the findings

The results are provided appropriately in detail.

Reviewer 2 ·

Basic reporting

The manuscript is generally clear and easy to understand. The English language could be further improved. The function of ticagrelor should be briefly introduced in the Abstract. The introduction could be further refined to emphasize the novelty and clinical value of the study. The article structure is ok. Raw data shared.

Experimental design

This is an animal study to evaluate the protective effect of ticagrelor of MIR-induced ALI. The authors found that pyroptosis-associated proteins including NLRP3, ASC, and cleaved caspase-1 were elevated in MIRI, and eliminated by ticagrelor. The study is interesting and reveals partly the underlying mechanism for MIR-induced ALI. There are some comments for the improvement of this study as follows:

1. Has there been any previous publication about this animal model of MIRI-induced ALI? If yes, please provide references.

2. In the animal model, ticagrelor was given before MIRI surgery, which was inconsistent with the clinical setting. Why not choose to give ticagrelor after the surgery since the time of active maximum concentration is only 2-3 hr?

3. Why all animals were sacrificed after 120 minutes? How do you confirm the drugs work within such a short observation time? I see in other references provided (line 262, Birnbaum et al. 2016), ticagrelor 150mg/kg daily was give for 3 days following MIRI. So how do you determine the observation time? Any previous experience for it?

4. One limitation is that survival is not evaluated in this study.

Validity of the findings

1.Please provide more evidence of the MIRI-induced ALI model.
2.The effect of ticagrelor I speculate is systemic. So it’s interesting to see if it also provide protective effect for the heart.

·

Basic reporting

Ticagrelor alleviates pyroptosis in myocardial ischemia reperfusion-induced acute lung injury.
Yi-Ning Dai 1, 2 , Li-Tao Wang 1, 2 , Ye-Shen Zhang 1, 2 , Ling Xue 1, 2 , Peng-Cheng He 1, 2 , Ning Tan 1, 2 , Yuan-Hui Liu
Liu et al present an interesting topic, where they trying to show that Ticagrelor alleviates pyroptosis in myocardial ischemia reperfusion-induced acute lung injury. Although the mechanism is not clear, this could have been a great advancement of our understanding the platelet aggregation and heart conditions. However, the authors have failed to perform comprehensive experimentation and analyses.

Experimental design

Broadly speaking authors have interesting ideas to work with however they have been significantly disappointed with the choice of experimental design. Given that they addressed the fundamental question, it could have been backed with different analyses where something like RNAseq, or FLOW analysis might have been a great addition. The number of replicates has not been mentioned either.

Validity of the findings

Reproducibility and validation are important, however, the data presented here do not look as if it has been replicated since they did not mention the replicates. Internal control b-actin does not make sense and legends need to be significantly improved.

Additional comments

It is important to pay a lot of attention to grammar and composition since there were paragraphs that did not make any sense whatsoever.

---

## Round 0.2 · Minor Revisions

The authors report that the Western blots were repeated (n=6) but they have only provided one example and so must upload all the uncropped blots for all Western blots rerported in the manuscript.

Reviewer 2 ·

Basic reporting

The authors have addressed my comments satisfactorily.

Experimental design

The authors have addressed my comments satisfactorily.

Validity of the findings

The authors have addressed my comments satisfactorily.

·

Basic reporting

Thank you very much for revising the manuscript. It’s in great position for publication. Hopefully this will help in advancing science as such, please keep up the great work, best wishes!

Experimental design

Thank you addressing the comments and revising the manuscript.

Validity of the findings

Manuscript and all the findings presented are vital for the publication. Thank you for great work!

Additional comments

Great job everyone, thank you for helping shape the science and our understanding!

---

## Round 0.3 · Major Revisions

I hope this message finds you well. The original Academic Editor is no longer available and so I have taken over the handling of your submission. After a very short and quick review of the submitted manuscript, I would like to provide some editorial suggestions to improve clarity and enhance the overall quality of your content.

1)There are several instances of language issues throughout the manuscript, such as the absence of a verb in lines 72-73 and the phrasing concerns in lines 76-79. To enhance readability and comprehension, I recommend addressing these & be throughout in the proofreading.

2) While the manuscript briefly touches on pyroptosis as a form of programmed cell death, it lacks a more comprehensive explanation of how pyroptosis distinguishes itself from other cellular processes. Additionally, the connection between pyroptosis and lung injury could be further elucidated to provide a clearer context.

3) The authors should emphasize the importance of understanding the proposed mechanism and clearly articulate the significance of their research. It would be beneficial to elaborate on why investigating this mechanism is relevant, especially in the context of AMI patients undergoing PCI.

4) To bolster the abstract's impact, I recommend incorporating quantitative data or statistical information. Additionally, it would be prudent for the authors to provide a detailed description of their densitometry methodology for Western blots (WBs) and immunohistochemical analyses (IHCs) in the Methods section. - do not forget to submit raw data and be transparent in this endeavor!
4.1) A careful evaluation of Figure 2 reveals concerns about the clarity of certain bands, particularly the 12KDa line. Furthermore, the pixelation observed in NLRP3 and caspase-1 bands is noteworthy. I suggest that the authors clarify the quantification process for WBs and IHCs to ensure reproducibility and accurate interpretation of results. Additionally, in Figure 1d, the absence of data points and the uneven distribution of dots among the groups should be addressed to enhance visual representation.

Finally, 5) Given the preliminary nature of the work, a more suitable title could better convey the scope and findings of the study. I encourage the authors to reconsider and revise the title accordingly.

Your attention to these points would greatly contribute to the manuscript's clarity, impact, and overall quality.

---

## Round 0.4 · Major Revisions

I repeat my previous decision. The authors completely ignored all of my previous remarks in the "comment to the author". I hope it was an oversight and not on purpose.

1) There are several instances of language issues throughout the manuscript, such as the absence of a verb in lines 72-73 and the phrasing concerns in lines 76-79. To enhance readability and comprehension, I recommend addressing these & be throughout in the proofreading.

2) While the manuscript briefly touches on pyroptosis as a form of programmed cell death, it lacks a more comprehensive explanation of how pyroptosis distinguishes itself from other cellular processes. Additionally, the connection between pyroptosis and lung injury could be further elucidated to provide a clearer context.

3) The authors should emphasize the importance of understanding the proposed mechanism and clearly articulate the significance of their research. It would be beneficial to elaborate on why investigating this mechanism is relevant, especially in the context of AMI patients undergoing PCI.

4) To bolster the abstract's impact, I recommend incorporating quantitative data or statistical information. Additionally, it would be prudent for the authors to provide a detailed description of their densitometry methodology for Western blots (WBs) and immunohistochemical analyses (IHCs) in the Methods section. - do not forget to submit raw data and be transparent in this endeavor!
4.1) A careful evaluation of Figure 2 reveals concerns about the clarity of certain bands, particularly the 12KDa line. Furthermore, the pixelation observed in NLRP3 and caspase-1 bands is noteworthy. I suggest that the authors clarify the quantification process for WBs and IHCs to ensure reproducibility and accurate interpretation of results. Additionally, in Figure 1d, the absence of data points and the uneven distribution of dots among the groups should be addressed to enhance visual representation.

Finally, 5) Given the preliminary nature of the work, a more suitable title could better convey the scope and findings of the study. I encourage the authors to reconsider and revise the title accordingly.

---

## Round 0.5 · Minor Revisions

Dear authors, thank you for your efforts and re-submission. Unfortunately, your manuscript can not yet be accepted for publication:

1) Language remains a problem (just an example line 42 "but this effect is not been reported" is incorrect; it should be "but this effect has not been reported")

2) figures or images quality: although your methods' description and your work seems scientifically sound, the quality of the images submitted, including in the supplementary (raw data) should be improved or of higher quality/resolution. Not sure if when you extracted the images you used an inadequate file extension? can you go back to the machine's software and improve the figures? At the moment, WB bands look pixelated and blurred, which should not be. Histology images look ok, though. Then, please include in the manuscript's figures' legends major features that the reader should pay attention to when looking at the images - sinalise it in the image itself too (eg. "Legend: (a) -Alveoli (*); (b) -blood vessel (V -artery), bronchioles (Bq); (c) -blood vessel (V -artery), alveolus (*); (d) -bronchus, vein (arrow), connective tissue (double arrow)" - something like this). Additionally, remember that you should include important stats in you graphs (Fig. 2 & 3, for example, include "Note: *P<0.05 compared with the sham group and **P<0.05 compared with the sepsis group." but no * or ** appear in any of the graphs).

Note that figure 5 has been highly zoomed in. This is unacceptable. Improve image resolution!

These are "basic" questions that can still cause your manuscript to be rejected, if not addressed. Be throughout in your revisions!

---

## Round 0.6 · accepted · Accept

Dear authors, many thanks for your submission and relentless work. I am happy to let you know that I am now accepting your work for publication. Please, keep in mind that the office may still request figures that fit the quality requirements for publication. Thank you, once again.